# Removal and Adsorption Mechanisms of Phosphorus, Cd and Pb from Wastewater Conferred by Landfill Leachate Sludge-Derived Biochar

**Huiqin Zhang [1,2], Kexin Lu [1], Juan Zhang [1], Chao Ma [1], Zixian Wang [1] and Xiaofang Tian [1,2,*]**

[1]  Innovation Demonstration Base of Ecological Environment Geotechnical and Ecological Restoration of Rivers and Lakes, School of Civil Engineering, Architecture and Environment, Hubei University of Technology, Wuhan 430068, China; hqzhang@hbut.edu.cn (H.Z.); 102110833@hbut.edu.cn (K.L.); 102200856@hbut.edu.cn (J.Z.); 102210859@hbut.edu.cn (C.M.); hanlingsha0707@163.com (Z.W.)
[2]  National Engineering Research Center of Advanced Technology and Equipment for Water Environment Pollution Monitoring, Wuhan 430068, China
*   Correspondence: 20120015@hbut.edu.cn

**Abstract:** There is a high treatment cost and secondary pollution to the environment due to the high organic content and complex composition in landfill leachate sludge in refuse incineration power plants. Landfill leachate sludge-derived biochar (LLSDB) was prepared via pyrolysis in order to realize its resource utilization and remove pollutants from wastewater. The study focused on the removal of nutrients phosphorus and heavy metals (Cd(II) and Pb(II)) from wastewater through the adsorption process using LLSDB. The investigation also looked into the kinetics and thermodynamics of the adsorption process. It was found that the Freundlich–Temkin–Langmuir model was the best model for describing the initial concentration of total phosphorus, (TP) 0–1.0, 1.0–20, and 20–120 mg/L, respectively, while the Freundlich–Langmuir model was the best model for Cd(II) 100–500 mg/L, Pb(II) 500–5000 mg/L, respectively. Additionally, while the exothermic entropy reduction process for TP ($<1.0$ mg/L) was spontaneous, the endothermic entropy increment processes for TP ($\geq 1.0$ mg/L), Pb(II) and Cd(II) in wastewater increased with the adsorption temperature. It was inferred for the adsorption mechanism of LLSDB that the adsorption of low concentrations of TP, Cd(II) and Pb(II) from wastewater was mainly physical adsorption, following a linear distribution, while that of high concentrations was mainly chemical adsorption because of a series of chemical reactions; TP, Cd(II) and Pb(II) from wastewater were nicely adsorbed and removed by LLSDB600, which was an incredibly superior strategy for controlling waste with waste.

**Keywords:** landfill leachate sludge-derived biochar; phosphorus; heavy metals; adsorption; mechanism

## 1. Introduction

The treatment of landfill leachate comes at a high cost and is difficult due to its complex composition, so the sludge produced by landfill leachate is more complex in composition and more harmful to the environment, which is likely to cause secondary pollution under improper treatment. The previous research results mainly focused on undesirable leaching, and the resulting accumulation of toxic heavy metals in the environment was of concern during biochar application because of the potentially negative impacts on public and ecological health. Previous studies have shown that landfill leachate sludge-derived biochar (LLSDB) prepared via pyrolysis technology at 600 °C could minimize the release of heavy metals/metalloids into an aqueous environment during LLSDB implementation for wastewater treatment [1]. Leaching contents of Mn, Zn, Cu, Cd, Cr, As, Ni, and Pb ($<0.5$ mg/L) were all less than their respective maximum discharge concentration standards for the first priority pollutants in China [1]. There has been no research on the adsorption and removal effects of LLSDB on common pollutants in wastewater, such as phosphorus and heavy metals.

Phosphorus is an indispensable nutrient element in human production, life, and other biological growth and development processes [2]. However, the excessive use of fertilizers, industrial wastewater, and domestic sewage discharge leads to excessive phosphorus being discharged into surface water and groundwater in human daily production and life [3]. Excessive phosphorus can lead to the eutrophication of water bodies, posing great harm to ecosystems and causing the proliferation of micro-organisms. Eventually, it can lead to the death of a large number of micro-organisms and algae, causing the water body to turn black and odorous due to factors such as hypoxia. Therefore, the development of economical and efficient phosphate removal technologies remains an essential issue in water environment management.

Heavy metal pollutants refer to metals and their compounds with atomic densities greater than 5 g/cm$^3$, mainly including Cu, Zn, Hg, Cd, Pb, Sn, Mn, and Cr, which were commonly considered the most common toxic pollutants in water and soil systems [4]. Heavy metal pollutants come from a wide range of sources (e.g., mining, industrial production and research, atmospheric sedimentation, exhaust emissions, fertilizers and pesticides used, material manufacturing, and wastewater discharge) and are distributed globally. It was found that heavy metals could persist in the body for a long time due to their non-biodegradability in the environment, continuously affecting various major organs such as the liver, heart, brain, and kidneys over time [5], disrupting their normal biological mechanisms, disrupting natural physiological responses, and posing long-term health risks [6], which were carcinogenic at low concentrations [7,8]. Especially heavy metals such as Cd and Pb belong to the first category of pollutants in wastewater discharge standards, which have become the most widely distributed toxic metals in the world [9]. Pb was a potential toxin that could disrupt the normal functioning of the body's biochemical and physiological processes. It was shown that the liver was firstly affected by ingested or inhaled Pb, leading to ultrastructural changes such as endoplasmic reticulum and mitochondria in organelle caused by Pb-induced hepatotoxicity [10]. When Pb interacts with the thiol groups of proteins in organisms, this damages the central nervous system of the human body, affecting the release of neurotransmitters, and causing diseases such as hepatitis, nephrotic syndrome and brain injury [11,12]. Cd is an essential toxicological metal that can damage important organs of the body after entering the human body [13] which accumulates in the liver, inducing lipid peroxidation in the body, and causing oxidative damage to the body, while also accumulating in the kidneys, interfering with the metabolism of Ca in the body, and even leading to renal failure and severe anemia. In addition, there was a significant toxic effect on the lungs, increasing the risk of cancer, cardiovascular disease and additional diseases [14].

It was necessary to strictly control the phenomenon of eutrophication in water bodies and reduce the risk of heavy metal release into the living environment, and to vigorously seek safer, environmentally friendly, and economic solutions. At present, effective methods for removing inorganic pollutants (such as phosphorus and heavy metals) from water bodies are mainly divided into three categories (physical–chemical, chemical and biological methods), including membrane separation, adsorption [15,16], precipitation [17], ion exchange [18], phytoremediation and biosorption [19,20], etc. The adsorption method has attracted widespread attention from researchers due to its characteristics of simple operation, high efficiency, economy, environmental friendliness, and low risk of secondary pollution. There are many kinds of adsorbents, mainly including activated carbon, nano materials, industrial waste residue, minerals, plant biomass and biochar [21,22]. In recent years, biochar has been widely used in the treatment of water pollutants due to its rich pore structure and excellent adsorption effect, becoming an effective environmentally friendly material [9,23,24]. It could be a win–win measure to solve the problems of sludge and water environment via the removal of pollutants by LLSDB from wastewater to explore the ability, adsorption efficiency and adsorption mechanism of LLSDB to treat wastewater pollutants (e.g., inorganic phosphate, heavy metal Cd(II) and Pb(II)), which has ecological and economic benefits to realizing the effective utilization of landfill leachate sludge (re-

duction, stabilization, harmlessness and resource utilization) and the effective removal of wastewater environmental pollutants. The findings of this study encourage further investigation to comprehensively evaluate the application prospects of LLSDB; the pollutants from wastewater were nicely adsorbed and removed by LLSDB as an adsorbent, which was an incredibly superior strategy of controlling waste with waste.

## 2. Experimental Section

### 2.1. Preparation of Landfill Leachate Sludge Derived Biochar

Landfill leachate sludge was collected from Green Power Renewable Energy Co., Ltd., Hongshan District, Wuhan, China. Once collected, the sludge was initially air-dried to remove most of the wastewater and then oven-dried at 105 °C until a constant weight was reached. The sludge was then ground in a mortar and sifted through a 100-mesh sieve. Pyrolysis was carried out in a tube furnace (KJ–T1200-S6008LK1–s) under vacuum conditions (0.1 MPa) for 1 h at a heating rate of $10 \, °C \cdot min^{-1}$. Various peak temperatures were employed to prepare different landfill leachate sludge-derived biochars (LLSDBs) for investigating the effect of pyrolysis temperature. The LLSDBs were stored in an airtight bag in a desiccator until use.

### 2.2. Adsorption of Representative Contaminants

To further examine the practicability of LLSDB, we selected TP and heavy metals (such as Cd(II), Pb(II)) in wastewater as representative contaminants. The adsorption process was performed on a frozen-floor shaking table (ZD-85) employing the ammonium molybdate spectrophotometric method (GB/T 11893-1989). An amount of 100 mL of a certain concentration of phosphate solution (calculated as TP) was prepared to add a certain amount of LLSDB and set a certain adsorption temperature, adsorption time, and different initial pH to explore the optimal adsorption conditions and complete the isothermal adsorption test. Finally, the adsorption removal of TP was determined by measuring the absorption peak of the solution after filtration with a 0.45 μm membrane via dual beam UV-visible spectrophotometry (TU-1901, Beijing General Analysis Instrument Co., Ltd., Beijing, China) while the adsorption removals of Cd(II) and Pb(II) were measured using atomic absorption spectrophotometer (AA-6880, Shimadzu Co., Kyoto, Japan). All analytical pure drugs used in the experiment were purchased from China National Pharmaceutical Group Chemical Reagent Co., Ltd., Shanghai, China. All samples to be tested were filtered by a 0.45 μm filter membrane. All the experiments were run in triplicates at the minimum. The reported results represent the mean value of the replicated sample. The error bars in the figures are one standard deviation of these measurements.

### 2.3. Characterizations

The content of conventional elements such as C, H, N, S and O in LLSDBs was measured at different pyrolysis temperatures using an element analyzer (FlashSmart, Thermo Fisher Scientific Inc., Waltham, MA, USA, Shanghai, China), while leached metals were analyzed using inductively coupled plasma–mass spectroscopy (PQ-MS, Germany, Shanghai, China). The pH of the leachate was measured using a pH meter. The specific surface area and pore size distribution of LLSDBs were determined using a specific surface area and porosity analyzer (Autosorb iQ, Quantachrome Ins., Boynton Beach, FL, USA, Shanghai, China). An X-ray diffractometer (XRD, D8 Advance, Brook, Germany) was used to analyze phase composition and structure (lynxeye detector, 40 kV voltage, 30 mA current, Cu anode target material, K-$\alpha$ radiation, scanning speed of 0.1 s/step, and sampling interval of 0.019450 step). A scanning electron microscope (SEM, Zeiss Sigma, UK, Shenzhen, China) was employed to visualize the morphology of the LLSDBs, and the resolution of the device was 1.3 nm (20 kV)/2.8 nm (1 kV), the acceleration voltage was 0.1–30 kV, and the amplification was 12 ×–1000 k×.

## 3. Results and Discussion

### 3.1. Physicochemical Properties of LLSDBs

The yield, ash content, specific surface area, total pore and pH of the LLSDBs are shown in Table 1. When the pyrolysis temperature of landfill leachate sludge increased from 300 °C to 700 °C [25], the yields of LLSDB decreased from $85.7 \pm 2\%$ to $67.9 \pm 1\%$ while the ash content of LLSDBs increased from $51.3 \pm 2\%$ to $69.4 \pm 2\%$. The specific surface area of LLSDBs went up firstly to 59.4 m$^2$/g and down to 55.0 m$^2$/g, and the total pore soared sharply from 2.2 to 19.1 cm$^3$/g, and then achieved stability. The pH value of solid LLSDBs continuously increased from 9.1 to 12.2. Pyrolysis was conducive to an increase in the specific surface area and total pore volume of LLSDBs, and more active sites were exposed to increase the adsorption capacity of LLSDBs. However, an excessive pyrolysis temperature would lead to the collapse and fragmentation of the original pore structure of LLSDBs, forming a larger specific surface area, but the total pore volume of LLSDBs was relatively constant. Additionally, sludge produced can lead to a decrease in acidic functional groups and an increase in alkaline substances [26].

**Table 1.** Yield, ash, specific surface area, total pore and pH of LLSDB at different pyrolysis temperatures.

| LLSDBs | Yield/% | Ash/% | Specific Surface Area (m$^2$/g) | Total Pore (cm$^3$/g) | pH |
|---|---|---|---|---|---|
| LLSDB300 | $85.7 \pm 2$ | $51.3 \pm 2$ | 42.1 | 2.2 | 9.1 |
| LLSDB400 | $77.3 \pm 2$ | $55.8 \pm 1$ | 42.2 | 15.3 | 9.5 |
| LLSDB500 | $71.2 \pm 1$ | $62.8 \pm 2$ | 49.2 | 16.2 | 10.7 |
| LLSDB600 | $69.8 \pm 2$ | $61.5 \pm 1$ | 59.4 | 19.1 | 11.3 |
| LLSDB700 | $67.9 \pm 1$ | $69.4 \pm 2$ | 55.0 | 19.1 | 12.2 |

The content of conventional elements was also measured. It can be seen that the content of elements C, N, and H in LLSDBs was negatively correlated with the pyrolysis temperature, while the content of element S in LLSDBs was positively correlated with the pyrolysis temperature in Table 2. However, the degree of reduction was different for each element, and there was no obvious shift for element O. Compared with the research on the preparation of biochars from general sludge [27], the proportion of C in LLSDB was relatively low and showed a decreasing trend. Due to the complex composition of landfill leachate sludge, functional groups containing C and H break due to their inability to withstand elevated temperatures when the pyrolysis temperature increases, resulting in a decrease in the content of C and H. From Table 2, it can be seen that the content of other substances was high, and it was speculated that the alkaline mineral content in LLSDB was relatively high based on the ash content and pH in Table 1.

**Table 2.** The composition of the major elements of LLSDB/%.

| LLSDBs | C | N | H | S | O | Others |
|---|---|---|---|---|---|---|
| LLSDB300 | 17.90 | 1.87 | 1.68 | 0.07 | 23.03 | 55.45 |
| LLSDB400 | 16.30 | 0.98 | 1.02 | 0.15 | 23.55 | 58.00 |
| LLSDB500 | 13.01 | 0.59 | 0.39 | 0.31 | 22.46 | 66.24 |
| LLSDB600 | 13.74 | 0.50 | 0.28 | 0.73 | 22.20 | 62.55 |
| LLSDB700 | 12.65 | 0.28 | 0.24 | 0.63 | 21.61 | 64.94 |
| LLSDB800 | 10.79 | 0.00 | 0.32 | 0.81 | 18.16 | 69.92 |

From Tables 1 and 2, it can be seen that the LLSDB600 surface had large specific surface area with a rich pore structure, which was conducive to adsorption. The alkaline substances in LLSDB could also alter the pH of the solution, changing the presence of pollutants in water. At the same time, LLSDBs with high pyrolysis temperatures released extra $CO_2$, which underwent secondary reactions with carbon atoms in the LLSDBs, thereby opening pores that were closed and increasing porosity [28,29]. However, when the pyrolysis temperature was greater than 600 °C, the aggregation and accumulation of micropores reduced the

specific surface area of biochars and reduced thermal stability [30,31]. Therefore, LLSDB600 was selected for subsequent research.

### 3.2. Adsorption Effect under Different Conditions

It was found that the adsorption rate of pollutants (TP, Cd(II), and Pb(II)) showed upward trends as the pyrolysis temperature increased, reaching the optimal adsorption rates, which were 63.4%, 78.3%, and 71.3%, respectively, at 600 °C and then stabilized, except for a decrease in the adsorption rate of Pb(II), which decreased to 70.3% at 800 °C as shown in Figure 1a. The porosity, structure, and functional groups of LLSDB varied at different pyrolysis temperatures. The specific surface area of LLSDB increased as the pyrolysis temperature increased (Table 1), increasing the activation energy required for the development of micropores, leading to the formation of new microporous structures [32]. Therefore, LLSDB600 was chosen for subsequent experiments.

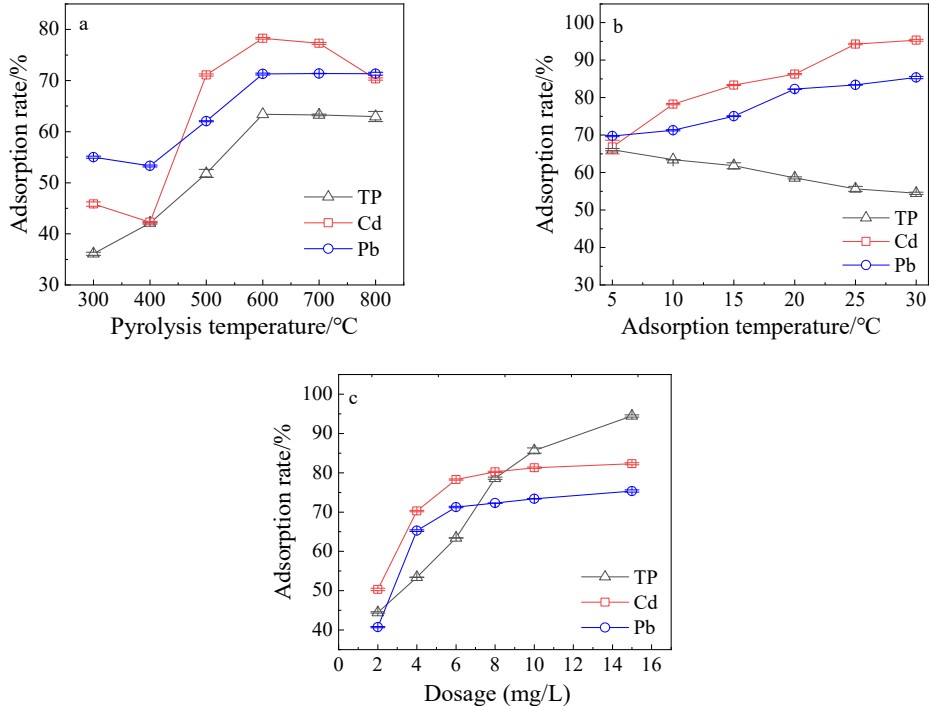

**Figure 1.** The adsorption effect of pollutants in wastewater under different conditions ((**a**) different pyrolysis temperatures of LLSDB with an adsorption temperature of 10 °C and dosage of 6.0 g/L; (**b**) different adsorption temperatures with the dosage of LLSDB600 being 6.0 g/L; (**c**) different dosages of LLSDB600 at an adsorption temperature of 10 °C; (**a**–**c**) initial concentrations of TP, Cd(II) and Pb(II) in wastewater, i.e., 0.7 mg/L, 200 mg/L and 1000 mg/L, respectively. Adsorption time of 2.0 h).

It was reflected that the adsorption rate of TP showed a downward trend from 66.1% to 54.5%, while the adsorption rates of Cd(II) and Pb(II) showed upward trends from 66.8%, 69.8% to 95.3%, 85.4%, respectively, when the adsorption temperature increased from 5 °C to 30 °C in Figure 1b. In general, the thermal movement of molecules accelerated when the adsorption temperature went up, which was not conducive to the adsorption process, while molecules were more likely to condense on the surface of the adsorbent due to a decrease in temperature. The lower the adsorption temperature is, the more it is favorable for physical adsorption. However, this might be the opposite case to that with chemical adsorption; Mishra et al. [33] thought that the adsorption performance of adsorbents increased with the increase in adsorption temperature because existing bonds were destroyed and new metal adsorption sites were formed.

It was revealed that the adsorption rate of TP, Cd(II) and Pb(II) showed upward trends from 44.5%, 50.3%, and 40.8% to 63.4%, 78.3%, and 72.3% when the dosage of

LLSDB600 increased from 2 g/L to 6 g/L as shown in Figure 1c; the adsorption rate of TP increased significantly as the dosage continued to increase, and reached 94.5% with dosage of 15 g/L, while the adsorption rate of Cd(II) and Pb(II) ascended slowly and reached stable levels. The total specific surface area and adsorption sites available for the adsorption of LLSDB600 increased when its dosage properly was increased, and thus the adsorption rate of pollutants was improved. Economically, it was sufficient for an appropriate dosage of 6 g/L to be used.

The concentration of $Ca^{2+}$ and pH in the solution changed slightly when the initial concentration of TP was low (e.g., 0.7 mg/L), but as the initial concentration of TP increased (>2.5 mg/L), the concentration of $Ca^{2+}$ in the solution gradually decreased from 105 mg/L to 0.5 mg/L, and the pH value of the solution also decreased from 11.3 to 10.8 as shown in Figure 2a. The concentration of $CO_3^{2-}$ and the pH the in solution changed slightly when the initial concentration of Cd(II) and Pb(II) was low, but as the initial concentration of Cd(II) and Pb(II) increased from 25 mg/L and 250 mg/L to 500 mg/L and 5000 mg/L, while the concentration of $CO_3^{2-}$ in solution gradually decreased from 150 mg/L to 0.5 mg/L and 15 mg/L, and the pH value of the solution also decreased from 11.2 to 9.7, 8.8, respectively, as shown in Figure 2b,c.

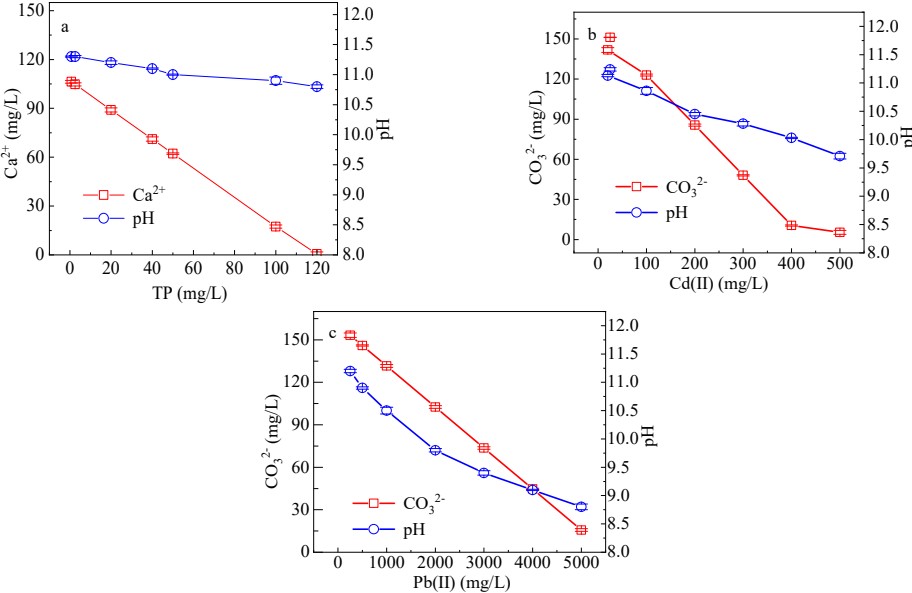

**Figure 2.** Changes in ion concentrations and pH in wastewater at different initial pollutant concentrations absorbed by LLSDB600 ((**a**) TP; (**b**) Cd(II); (**c**) Pb(II). LLSDB dosage of 6 g/L, adsorption temperature of 10 °C and adsorption time of 2.0 h).

### *3.3. Kinetics and the Thermodynamics for Adsorption*

### 3.3.1. Kinetics for Adsorption

The isothermal adsorption results were fitted using the Langmuir equation and Freundlich equation [34], which is shown in Figure 3. The adsorption isotherm of LLSDB600 on TP in wastewater was obtained with the adsorption temperature set to 5 °C, 15 °C and 25 °C. The lower the adsorption temperature, the better the adsorption effect when the initial concentration of TP was less than 0.9 mg/L (Figure 3a), and it was concluded that physical adsorption played a dominant role [35], but the adsorption rate of TP was not significant (only about 65%). On the other hand, the higher the adsorption temperature, the better the adsorption effect when the range of TP was 1–20 mg/L, the exponential growth was rapid with 20–60 mg/L, the growth rate gradually slowed down with 60–80 mg/L, the adsorption gradually became saturated as the concentration reached 80 mg/L, and the adsorption rates of TP for LLSDB600 reached the highest of 85.4% and 88.8%, respectively, when TP was 2.5 mg/L and 40 mg/L at 25 °C. The adsorption rate of TP increased because

the increase in surface porosity and active adsorption sites of LLSDB led to an increase in adsorption rate with the increase in TP concentration and adsorption temperature. It was beneficial for chemical adsorption to have a high adsorption temperature [36], and chemical adsorption dominated when TP was greater than 20 mg/L, while there was a decrease in adsorption rate of TP when TP was greater than 40 mg/L because of the saturated adsorption site and the excessive phosphate.

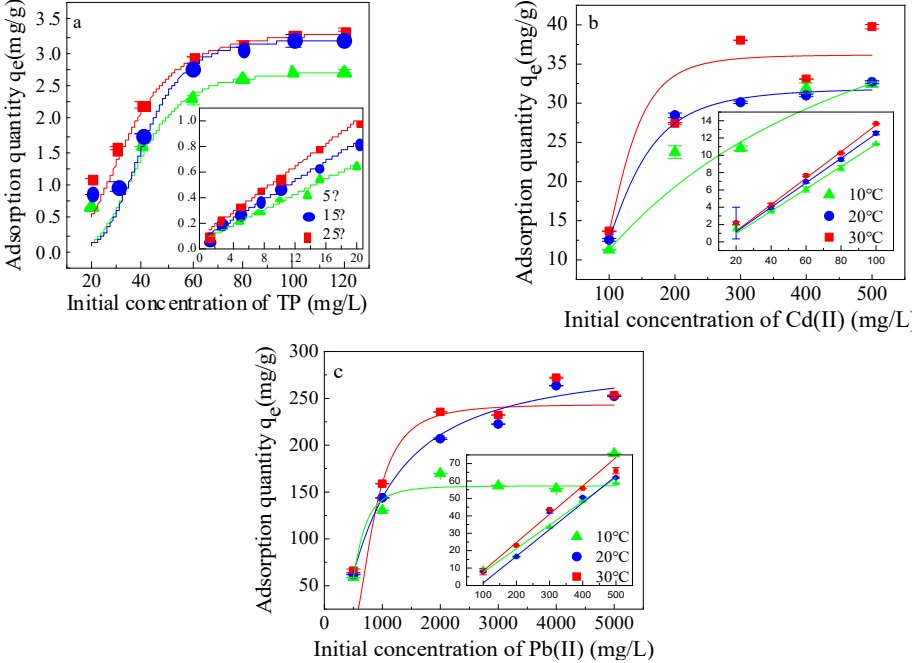

**Figure 3.** Adsorption isotherm of pollutants in wastewater ((**a**) initial concentration of TP; (**b**) initial concentration of Cd(II); (**c**) initial concentration of Pb(II). Adsorption time of 2 h, and dosage of 6 g/L).

The adsorption isotherms of LLSDB600 on Cd(II) and Pb(II) in wastewater were obtained with the adsorption temperature being set to 10 °C, 20 °C and 30 °C (Figure 3b,c). The higher the adsorption temperature, the better the adsorption effect of Cd(II) and Pb(II) in wastewater. The adsorption of Cd(II) and Pb(II) by LLSDB600 showed a linear increase in the initial concentration range of 0–100 mg/L and 0–500 mg/L, respectively, following linear patterns. It showed a rapid exponential growth for the adsorption of Cd(II) and Pb(II) with 100–200 mg/L and 500–2000 mg/L, respectively. It was found that the growth rates gradually slowed down in the ranges of 200–500 mg/L and 2000–5000 mg/L, and the adsorption gradually tended to be saturated with Cd(II) and Pb(II) reaching 300 mg/L and 4000 mg/L, respectively. It was shown that it followed linear distributions for the LLSDB600 adsorption of TP, Cd(II) and Pb(II) at low concentrations.

It was indicated that LLSDB600 adsorbed TP in wastewater mainly in a single layer while it adsorbed Cd(II) and Pb(II) in wastewater mainly in a multi-molecular layer at high concentrations. It was found that the adsorption effect on Pb(II) in wastewater was better than that on Cd(II) in wastewater because the adsorption capacity of LLSDB600 for Pb(II) in wastewater was considerably greater than that for Cd(II) in wastewater.

The adsorption isotherm was used to explore the relationship between the concentration and adsorption capacity at the adsorption equilibrium. The Langmuir model, Freundlich model, Dubin Radushkevich (D-R) model and Temkin model were used to fit and analyze the experimental data obtained, discus the saturated adsorption capacity and various fitting parameters under different adsorption temperatures and initial concentrations, and explore the adsorption mechanism, as shown in Table 3. For the low-concentration solution of TP (0–1.0 mg/L) at 5 °C, the adsorption followed the Freundlich model ($R^2$ = 0.96, *n* = 0.5). A larger $K_F$ value indicates a stronger adsorption capacity. The

adsorption capacity became stronger when the adsorption temperature was lower and the deviation between the adsorption isotherm and the linearity became smaller when the *n* value decreased. Therefore, it is speculated that the adsorption mechanism for TP in wastewater at low concentrations was physical adsorption dominated by a linear distribution [37,38]. The adsorption for TP in the concentration solution of 1.0–20 mg/L followed the Temkin model with $R^2$ from 0.86 to 0.95 when its adsorption temperature went up from 5 °C to 25 °C; the higher the adsorption temperature, the greater the adsorption amount, which refers to the linear decrease in the adsorption heat of the adsorbent on the surface with increasing coverage, describing the electrostatic interaction in the Temkin model [39]. The adsorption mechanism was mainly physical adsorption dominated by electrostatic interaction. The adsorption for TP in the concentration solution of 20–120 mg/L followed the Langmuir model with the value of $R^2$ being from 0.70 to 0.94 when its adsorption temperature went up from 5 °C to 25 °C; the higher the adsorption temperature, the better the fitting effect [40]. The adsorption for Cd(II) in the concentration solution of 100–500 mg/L followed the Langmuir model with the value of $R^2$ being from 0.95 and 0.97 to 0.99 while the adsorption for Pb(II) in the concentration solution of 500–5000 mg/L followed the Freundlich model with the value of $R^2$ being from 0.82 and 0.98 to 0.95 when its adsorption temperature went up from 10 °C, 20 °C to 30 °C; the higher the adsorption temperature, the greater the adsorption amount [41].

The separation factor was $0 < K_L < 1.0$, indicating that the higher the temperature, the better the adsorption effect [42], and the $K_L$ value tended to 0, which is in line with the irreversible characteristics of chemical adsorption [42]. At the same time, it can be seen from the Freundlich model parameters that the larger the n, the greater its linear deviation. The adsorption process was transformed from reversible adsorption to irreversible adsorption and the adsorption mechanism was irreversible, single-molecular-layer, chemical adsorption.

### 3.3.2. Thermodynamics for Adsorption

Gibbs free energy change ($\Delta G^0$), standard enthalpy change ($\Delta H^0$) and standard entropy change ($\Delta S^0$) were calculated and the thermodynamic properties of adsorption were obtained by using formulas to obtain the correlation coefficients, as shown in Table 4. It was indicated that the adsorption process of LLSDB600 on TP (0–1.0 mg/L) in wastewater was spontaneous ($\Delta G^0 < 0$), invoving an exothermic reaction ($\Delta H^0 < 0$) and a decrease in entropy ($\Delta S^0 < 0$), while the value of $\Delta G^0$ increased for the TP when the adsorption temperature ascended, indicating that it reduced the spontaneous trend of adsorption when the adsorption temperature increased, which was not conducive to adsorption, thus obtaining beneficial for adsorption at low concentrations and low temperatures, as is consistent with the previous analysis. It was indicated that the adsorption process of LLSDB600 on TP ($\geq$1.0 mg/L), Cd(II) and Pb(II) in wastewater was spontaneous ($\Delta G^0 < 0$), involving an endothermic reaction ($\Delta H^0 > 0$) and an increase in entropy ($\Delta S^0 > 0$), while the value of $\Delta G^0$ decreased when the adsorption temperature ascended, indicating that it added the spontaneous trend of adsorption when the adsorption temperature increased, which was conducive to adsorption; the higher the adsorption temperature, the stronger the spontaneous adsorption ability in the adsorption process of LLSDB600 on TP ($\geq$1.0 mg/L), Cd(II) and Pb(II) in wastewater [43,44]. Therefore, it was a spontaneous, exothermic, entropy reduction process for TP (<1.0 mg/L) while it was spontaneous, endothermic, entropy increment processes for TP ($\geq$1.0 mg/L), Cd(II) and Pb(II) adsorbed by LLSDB600 when the adsorption temperature increased.

**Table 3.** Fitting parameters of adsorption isotherm in different adsorption models.

| Model | Parameter | Adsorption Temperature (°C) and Initial Concentration (mg/L) for TP | | | | | | | | |
|---|---|---|---|---|---|---|---|---|---|---|
| | | 5 °C | | | 15 °C | | | 25 °C | | |
| | | 0–1.0 | 1.0–20 | 20–120 | 0–1.0 | 1.0–20 | 20–120 | 0–1.0 | 1.0–20 | 20–120 |
| Langmuir $q_e = \frac{q_m K_L c_e}{1 + K_L c_e}$ | $q_m$ (mg/g) | 0.02 | 0.88 | 4.55 | 0.04 | 1.16 | 4.76 | 0.02 | 1.75 | 4.35 |
| | $K_L$ (L/mg) | −2.15 | 0.20 | 0.03 | −1.17 | 0.25 | 0.05 | −1.54 | 0.24 | 0.10 |
| | $R^2$ | 0.77 | 0.83 | 0.70 | 0.29 | 0.80 | 0.81 | 0.28 | 0.51 | 0.94 |
| Freundlich $q_e = K_F \lg C_e^{1/n}$ | $K_F$ (L/mg) | 0.44 | 0.15 | 0.60 | 0.42 | 0.22 | 0.98 | 0.04 | 0.33 | 1.35 |
| | $1/n$ | 1.99 | 0.61 | 0.39 | 2.20 | 0.67 | 0.32 | 0.56 | 0.77 | 0.25 |
| | $R^2$ | 0.96 | 0.87 | 0.49 | 0.39 | 0.87 | 0.36 | −0.31 | 0.85 | 0.40 |
| D-R $q_e = \frac{q_m}{(1 + 1/C_e)^{\beta R^2 T^2}}$ | $\beta$ | 1.74 | 2.16 | 8.75 | 2.17 | 1.93 | 3.10 | 1.53 | 1.69 | 9.48 |
| | $q_m$ (mg/g) | 0.48 | 0.68 | 2.88 | 0.47 | 0.78 | 3.24 | 0.23 | 0.89 | 3.16 |
| | E | 0.51 | 0.49 | 0.31 | 0.49 | 0.50 | 0.44 | 0.53 | 0.52 | 0.30 |
| Temkin $q_e = A \ln K_t C_e$ | $K_t$ | 2.44 | 0.64 | 0.33 | 2.41 | 0.70 | 0.64 | 0.77 | 0.75 | 1.34 |
| | A | 0.11 | 0.33 | 2.25 | 0.09 | 0.45 | 2.33 | −0.03 | 0.61 | 1.96 |
| | $R^2$ | 0.91 | 0.86 | 0.69 | 0.09 | 0.93 | 0.58 | −0.30 | 0.95 | 0.61 |

| Model | Parameter | Adsorption temperature (°C) and initial concentration (mg/L) for Cd(II)/Pb(II) | | | | | |
|---|---|---|---|---|---|---|---|
| | | 10 °C | | 20 °C | | 30 °C | |
| | | 0–100/0–500 | 100–500/500–5000 | 0–100/0–500 | 100–500/500–5000 | 0–100/0–500 | 100–500/500–5000 |
| Langmuir $q_e = \frac{q_m K_L c_e}{1 + K_L c_e}$ | $q_m$ (mg/g) | 2.27/58.78 | 181.82/36.90 | 8.13/85.47 | 33.44/263.16 | 30.67/185.19 | 38.76/263.16 |
| | $K_L$ (L/mg) | 0.001/0.013 | 0.006/0.021 | 1.30/0.007 | 0.091/0.006 | 0.022/0.003 | 0.10/0.017 |
| | $R^2$ | 0.02/0.21 | 0.83/0.95 | 0.51/0.71 | 0.97/0.49 | 0.59/0.42 | 0.99/0.77 |
| Freundlich $q_e = K_F \lg C_e^{1/n}$ | $K_F$ (L/mg) | 0.39/3.68 | 10.30/81.47 | 2.02/0.95 | 21.48/58.88 | 0.93/0.34 | 20.50/104.71 |
| | $1/n$ | 0.89/0.48 | 0.20/0.092 | 0.45/0.81 | 0.10/0.18 | 0.79/1.15 | 0.007/0.11 |
| | $R^2$ | 0.89/0.68 | 0.69/0.82 | 0.73/0.51 | 0.47/0.98 | 0.76/0.68 | 0.98/0.93 |
| D-R $q_e = \frac{q_m}{(1 + 1/C_e)^{\beta R^2 T^2}}$ | $\beta$ | 2.69/4.55 | 66.21/0.001 | 0.12/54.64 | 13.21/0.001 | 4143.81/252,068.6 | 69,636.61/296,844.2 |
| | $q_m$ (mg/g) | 2.35/4.72 | 4.39/1.00 | 2.41/5.14 | 4.47/1.00 | 2.56/5.77 | 4.82/10.28 |
| | E | 0.46/0.39 | 0.12/0.71 | 0.69/0.14 | 0.26/0.71 | 0.016/5.77 | 0.004/0.002 |
| Temkin $q_e = A \ln K_t C_e$ | $K_t$ | 0.57/0.59 | 1.04/5.77 | 1.35/0.36 | 11.67/0.68 | 0.76/0.32 | 37.36/3.82 |
| | A | 8.94/26.20 | 12.61/32.74 | 5.21/45.63 | 7.79/82.33 | 9.75/70.92 | 5.29/54.18 |
| | $R^2$ | 0.69/0.51 | 0.69/0.55 | 0.54/0.43 | 0.44/0.95 | 0.56/0.64 | 0.97/0.89 |

Note: $qe$ in the equation is the equilibrium adsorption capacity (mg/g), and $K_L$ is the Langmuir constant (L/mg); $K_F$ (L/mg) and n (dimensionless) are Freundlich isothermal constants; $\beta$ is the adsorption's free energy activity, R is the standard molar constant, $8.314 \times 10^{-3}$ KJ/(mol·K); A and $Kt$ are constants of the Temkin isothermal model.

**Table 4.** Thermodynamic parameters.

| Concentration (mg/L) | | Temperature (°C) | $\Delta G^0$ (kJ/mol) | $\Delta H^0$ (kJ/mol) | $\Delta S^0$ (kJ/mol·K) |
|---|---|---|---|---|---|
| TP | 0–1 | 5 | −1.06 | −237.30 | −0.85 |
| | | 15 | −0.98 | | |
| | | 25 | −0.95 | | |
| | 1–20 | 5 | −0.11 | 1986.34 | 7.14 |
| | | 15 | −0.61 | | |
| | | 25 | −1.59 | | |
| | 20–120 | 5 | −0.08 | 1851.31 | 6.66 |
| | | 15 | −0.75 | | |
| | | 25 | −0.93 | | |
| Cd(II) | 0–100 | 10 | −0.28 | 16.82 | 0.060 |
| | | 20 | −0.88 | | |
| | | 30 | −1.44 | | |
| | 100–500 | 10 | −2.14 | 43.18 | 0.16 |
| | | 20 | −3.74 | | |
| | | 30 | −4.44 | | |
| Pb(II) | 0–500 | 10 | −0.015 | 2.19 | 0.00 |
| | | 20 | −0.021 | | |
| | | 30 | −2.01 | | |
| | 500–5000 | 10 | −3.02 | 38.3 | 0.15 |
| | | 20 | −4.48 | | |
| | | 30 | −7.59 | | |

Note: $\Delta G^0 = -RT \ln K_d$; $\ln k_d = \frac{\Delta S^0}{R} - \frac{\Delta H^0}{RT}$; $K_d = \frac{C_a}{C_e}$. $K_d$ is the adsorption equilibrium constant, and Ca is the concentration adsorbed by adsorbate (mg/L); $T$ (K) represents thermodynamic temperature; $R$ is the standard molar constant, $8.314 \times 10^{-3}$ KJ/(mol·K).

### 3.4. Adsorption Characteristics of LLSDB600

The XRD analysis of LLSDB600-adsorbed TP, Cd(II) and Pb(II), respectively, in wastewater is shown in Figure 4a–e. The strong diffraction peaks of $CaCO_3$, as well as those of $K_2O$, $(Mg_{0.03}Ca_{0.97})$ $(CO_3)$, and Ca $(OH)_2$, indicated the presence of K, Ca, and Mg in LLSDB600, as seen in Figure 4a. There also were main characteristic peaks of $CaCO_3$, $(Mg_{0.03}Ca_{0.97})$ $(CO_3)$ as shown in Figure 4b, but the peaks of $K_2O$ and $Ca(OH)_2$ disappeared, and no obvious peak related to phosphorus, as shown in Figure 4b. The main characteristic peak of $CaCO_3$ still existed and two new diffraction peaks of $CaPO_3(OH)\cdot2H_2O$ and $CaHPO_4(H_2O)_2$ were observed, as shown in Figure 4c. It was found that the diffraction peaks of $CaCO_3$ and $(Mg_{0.03}Ca_{0.97})(CO_3)$ still existed and that the original peak of $CdCO_3$ was generated in Figure 4d, and new peak $PbCO_3$ was generated in Figure 4e, which was consistent with that obtained via the analysis shown in Figure 2, as chemical adsorption occurred and precipitates were generated, resulting in a decrease in pH and ion concentrations such as those of $Ca^{2+}$ and $CO_3^{2-}$ in the solution. The apparent morphologies of LLSDB600 before and after the adsorption of TP, Cd(II) and Pb(II) are shown in Figure 4I–IV. Before adsorption (Figure 4I), there were many uneven pores on the surface of LLSDB600, and the pore structure was relatively developed, with a large specific surface area. An obvious carbon structure could be seen and a higher specific surface area was more conducive to adsorbing pollutants, with sufficient adsorption sites. After adsorption, a large number of small particle clusters gathered on the surface of LLSDB600, filling most of the pores that originally existed on the surface, and there was a large number of small particles arranged densely on LLSDB600′s surface, embedded in the microporous structure. Solid particles may have been generated by calcium phosphate, hydroxides, or carbonate deposited on the surface of LLSDB600 (as indicated by green circles in Figure 4II–IV). It was further confirmed that LLSDB600 adsorbed the high concentrations of TP, Cd(II) and

Pb(II) from wastewater, and chemical adsorption occurred, as determined through XRD and SEM analysis.

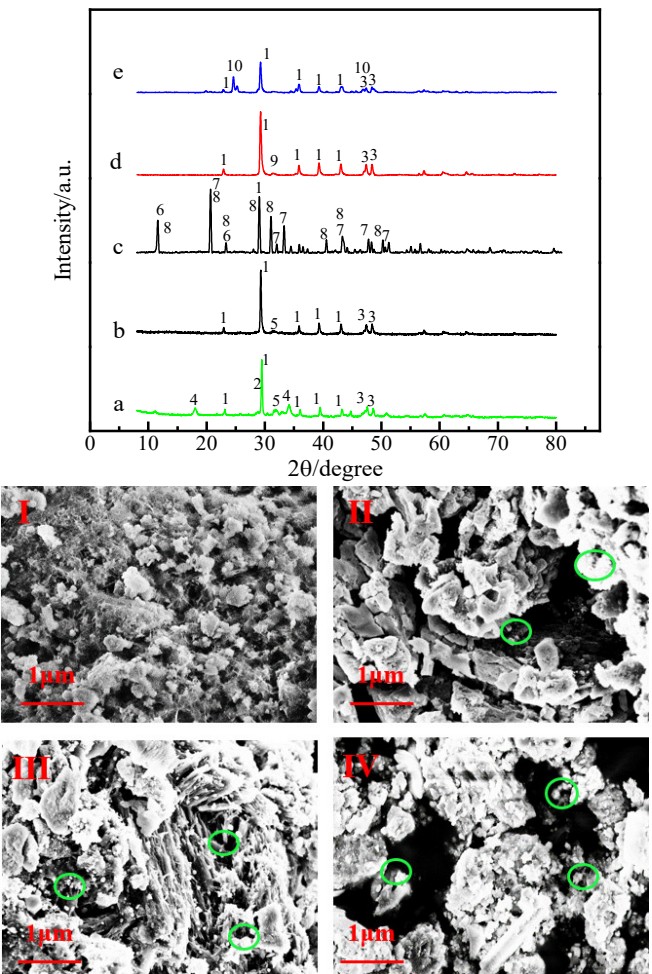

**Figure 4.** XRD patterns and SEM images of LLSDB (XRD—(**a**): LLSDB600, (**b**): LLSDB600 adsorbed 0.7 mg/L TP, (**c**): LLSDB600 adsorbed 100 mg/L TP, (**d**): LLSDB600 adsorbed 200 mg/L Cd(II) of, (**e**): LLSDB600 adsorbed 1000 mg/L Pb(II). 1—$CaCO_3$, 2—$K_2O$, 3—$(Mg_{0.03}Ca_{0.97})(CO_3)$, 4—$Ca(OH)_2$, 5—$NaCl$, 6—$Al_2Mg_4(OH)_{12}(CO_3)(H_2O)_3$, 7—$CaPO_3(OH)\cdot2H_2O$, 8—$CaHPO_4(H_2O)_2$, 9—$CdCO_3$, and 10—$PbCO_3$. Si and other elements not marked. SEM—(**I**): LLSDB600, (**II**): LLSDB600 adsorbed TP, (**III**): LLSDB600 adsorbed Cd(II), (**IV**): LLSDB600 adsorbed Pb(II). The green circles indicated the small particles after adsorption).

The module of HYDRA was the hydrochemical equilibrium constant database, which provided various chemical reaction equilibrium constants of thousands of chemical species of 166 common basic chemical components. Its characteristics were simple, open and flexible. The module MEDUSA is a module for calculation and mapping based on the chemical equilibrium system determined by HYDRA [45]. The morphology distribution of the equilibrium system can be obtained by setting balance conditions and parameter ranges, selecting mapping types. The software HYDRA/MEDUSA was very simple and intuitive to use to plot the morphological distribution of the chemical equilibrium system, which makes easy to master and expands its applications [45,46]. The morphological composition of each substance was calculated and drawn in the equilibrium system using the HYDRA/MEDUSA software when the pollutants with different initial pH values. The initial concentrations of TP, Cd(II) and Pb(II) in wastewater were adsorbed by LLSDB600, and the concentration of $Ca^{2+}$ and $CO_3^{2-}$ was 107 mg/L and 160.5 mg/L, respectively, in LLSDB600 as shown in Figure 2 [31] and Figure 5. It can be seen that there was no

precipitation generated and that the adsorption mechanism was physical adsorption when phosphate of a low concentration (0.7 mg/L) in wastewater was adsorbed by LLSDB600; there was a modest amount of precipitation of $Ca_5(PO_4)_3OH$ generated when the phosphate concentration increased to 2.5 mg/L, while there was a large amount of precipitation of $Ca_5(PO_4)_3OH$ generated when the phosphate concentration was ≥40 mg/L. It can be found that the main precipitates were $CdCO_3$ and $Cd(OH)_2$ when the initial concentration of Cd(II) was ≥80 mg/L; however, the precipitate of $Cd(OH)_2$ generated required the pH in wastewater to reach a value above 11.0 (Figure 5e–f). It can be seen that the main precipitates were $PbCO_3$ and $Pb(OH)_2$ when Pb(II) with the initial concentration of 200 mg/L in wastewater was adsorbed by LLSDB600, while some of the precipitate of $PbCO_3$ was converted into $Pb(OH)_2$ as the initial concentration of Pb(II) increased to 1000 mg/L (Figure 5g–h).

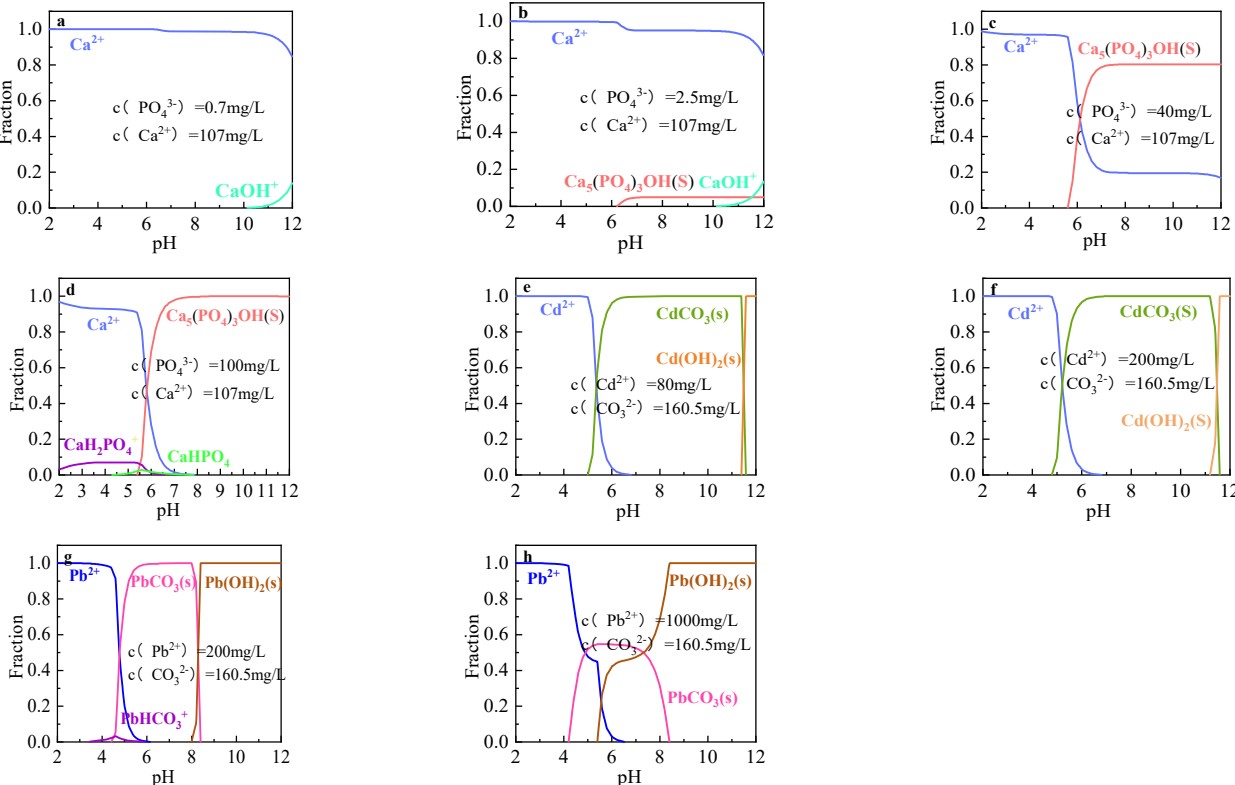

**Figure 5.** Precipitation dissolution equilibrium of LLSDB600 adsorbed pollutants in wastewater under different pH conditions ((**a**–**d**) initial concentrations of TP, 0.7, 2.5, 40 and 100 mg/L, respectively; (**e**,**f**) initial concentrations of Cd(II), 80 and 200 mg/L, respectively; (**g**,**h**) initial concentrations of Pb(II), 200 and 1000 mg/L, respectively).

In the above XRD detection analysis, no $Cd(OH)_2$ and $Pb(OH)_2$ was found. The reason discovered was that $Cd(OH)_2$ and $Pb(OH)_2$ were prone to reacting with $CO_2$ or $CO_3^{2-}$ in the wastewater, and converted into $CdCO_3$ and $PbCO_3$, respectively.

### 3.5. Proposed Adsorbed Mechanisms

Based on the above analysis, physical adsorption was dominant under a low phosphate concentration while chemical adsorption was dominant under a high phosphate concentration. There were more alkaline substances such as $Ca(OH)_2$, $K_2O$, $CaCO_3$ and $(Mg_{0.03}Ca_{0.97})(CO_3)$ in LLSDB600, which dissolved in wastewater and reacted to ionize $Ca^{2+}$, $Mg^{2+}$, $K^+$, $CO_3^{2-}$, and $OH^-$ [47].

Phosphate ions mainly existed in the form of $H_2PO_4^-$ when the pH in solution was low (<3.0). The pH in the wastewater was increased (>8.0) by $OH^-$ after the addition of

LLSDB600 (Table 1, pH = 11.3) and $OH^-$ underwent a chemical reaction with $H_2PO_4^-$; thus, the existence state of phosphate ions gradually changed from $H_2PO_4^-$ to $HPO_4^{2-}$ and $PO_4^{3-}$. The solubility of $HPO_4^{2-}$ and $PO_4^{3-}$ in the wastewater was easier to destabilize than that of $H_2PO_4^-$, while $Ca^{2+}$ reacted with $HPO_4^{2-}$ and $PO_4^{3-}$ in wastewater to generate substances such as calcium phosphate (e.g., $CaPO_3(OH) \cdot 2H_2O$, $CaHPO_4(H_2O)_2$) during the process of phosphate adsorption by LLSDB600, and they were enriched and adsorbed on the surface of LLSDB600 in the form of precipitation [48] but precipitates such as calcium phosphate could be generated only at a high initial concentration of phosphate while they could not be generated at low concentrations. $OH^-$ reacted with Cd(II) and Pb(II) in the wastewater to generate the substances $Cd(OH)_2$, $Pb(OH)_2$, and $Cd(OH)_2$; $Pb(OH)_2$ was also prone to react with $CO_2$ in the air to generate $CdCO_3$, $PbCO_3$, and Cd(II), and Pb(II) could also react with $CO_3^{2-}$ ($CaCO_3$ ionized to generate $Ca^{2+}$ and $CO_3^{2-}$ in the wastewater) to generate $CdCO_3$ and $PbCO_3$. The mainly chemical reactions associated were as follows [31,47]:

(1) The pyrolysis process and decomposition for LLSDB:

$$Ca(OH)_2(s) \overset{heat}{\to} CaO(s) + H_2O(g) \tag{1}$$

$$CaCO_3, K_2CO_3(s) \overset{heat}{\to} CaO, K_2O(s) + CO_2(g) \tag{2}$$

(2) Dissolution for LLSDB and pH increase in the wastewater:

$$CaO, K_2O(s) + H_2O(l) \overset{water}{\to} Ca(OH)_2, KOH(s) \tag{3}$$

$$Ca(OH)_2, KOH(s) \overset{water}{\leftrightarrow} Ca^{2+}, K^+ + OH^- \tag{4}$$

$$CaCO_3, K_2CO_3(s) \overset{water}{\leftrightarrow} Ca^{2+}, K^+ + CO_3^{2-} \tag{5}$$

(3) Precipitation for pollutions by LLSDB in the wastewater:

$$OH^- + H_2PO_4^- \overset{water}{\to} HPO_4^{2-}, PO_4^{3-} \tag{6}$$

$$Ca^{2+} + HPO_4^{2-}, PO_4^{3-} + OH^- + H_2O \overset{water}{\to} CaPO_3(OH) \cdot 2H_2O, CaHPO_4(H_2O)_2(s) \tag{7}$$

$$Cd^{2+}, Pb^{2+} + OH^- \overset{water}{\to} Cd(OH)_2, Pb(OH)_2(s) \tag{8}$$

$$Cd(OH)_2, Pb(OH)_2(s) + CO_2 \overset{water}{\to} CdCO_3(s), PbCO_3 + H_2O \tag{9}$$

$$Cd^{2+}, Pb^{2+} + CO_3^{2-} \overset{water}{\to} CdCO_3, PbCO_3(s) \tag{10}$$

(4) Exchange for pollution by LLSDB in the wastewater:

$$CaPO_3(OH) \cdot 2H_2O, CaHPO_4(H_2O)_2(s) + CO_3^{2-} \overset{water}{\leftrightarrow} CaCO_3(s) + HPO_4^{2-}, PO_4^{3-} + H_2O \tag{11}$$

$$Cd^{2+}, Pb^{2+} + R - Ca^{2+} \overset{water}{\leftrightarrow} R - Cd^{2+}, Pb^{2+} + Ca^{2+} \tag{12}$$

## 4. Conclusions

Biochar technology has ecological, social and economic benefits in many aspects and dimensions. The landfill leachate sludge had a high organic content and complex composition, and contained different kinds of heavy metals, which needed to be solidified and treated as hazardous waste, led to a high treatment cost and easily created secondary pollution to the environment. The sludge was prepared into LLSDB in order to achieve its reduction and resource utilization; it was found that LLSDB600 could minimize the release of heavy metals/metalloids into an aqueous environment during the implementation of LLSDB for wastewater treatment, and also ensured its harmlessness. Even more

impressively, the nutrients TP and heavy metals Cd and Pb in the wastewater were well adsorbed and removed by LLSDB600, which allowed the realization of its resource utilization. It was shown that there was a spontaneous, exothermic, entropy reduction process for TP(<1.0 mg/L) while there was a spontaneous, endothermic, entropy increment processes for TP($\geq$1.0 mg/L), Cd(II) and Pb(II) in the adsorption process of LLSDB600 on TP in wastewater, which followed linear distributions for the LLSDB600 adsorption of TP, Cd(II) and Pb(II) at low concentrations, which was dominated by physical adsorption. LLSDB600 adsorbed TP in the wastewater mainly in a single layer while LLSDB600 adsorbed Cd(II) and Pb(II) in wastewater mainly in a multi−molecular layer at high concentrations, a process which was dominated by chemical adsorption, so this indicates that low temperatures and concentrations are conducive to physical adsorption, while high temperatures and concentrations are conducive to chemical adsorption. In addition, the adsorption capacity for Pb(II) in the wastewater was better than that for Cd(II) in the wastewater. Additionally, LLSDB600−adsorbed pollutants can be treated as hazardous waste, requiring treatment with low treatment costs and a low risk of secondary pollution, obtaining the purposes of achieving a reduction in consumption, a decrease in pollution and an increase in benefits to a large extent. The findings of this study encourage further investigation to comprehensively achieve the target of changing waste matters into useful materials.

**Author Contributions:** Writing—reviewing & editing, H.Z.; Data curation, K.L.; Software, J.Z.; Investigation, C.M.; Resources, Z.W.; Supervision, X.T. All authors have read and agreed to the published version of the manuscript.

**Funding:** This project was supported by the National Natural Science Foundation of China (51808202) and Project from Hubei University of Technology (2018014).

**Institutional Review Board Statement:** Not applicable.

**Informed Consent Statement:** Not applicable.

**Data Availability Statement:** Not applicable.

**Conflicts of Interest:** The authors declare no conflict of interest.

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
