# Peer review of "Removal and Adsorption Mechanisms of Phosphorus, Cd and Pb from Wastewater Conferred by Landfill Leachate Sludge-Derived Biochar"

_sustainability, doi:10.3390/su151310045_

Round 1
Reviewer 1 Report
The submitted paper reports the adsorption of P, Pb and Cd on landfill leachate sludge-derived biochar. The adsorption mechanisms are discussed. It was proved that the removal of cations was mainly controlled by the mineralization induced by the released chemicals. However, at low concentration, it is said the uptake was mainly through physical adsorption, which was based on the results of simple modeling. I do not think it is accurate and convincing. The authors should setup more experiments and provide more proofs. Moreover, as regard to the mineralization, more details should be given, such as the change of pH and released ions in solution. Therefore, I recommend it to be resubmitted after a major revision.
Moderate modification is needed.
Author Response
We are very grateful to Reviewer for reviewing the paper so carefully.
We have completely revised the manuscript to clarify the points raised by the reviewer(s). The responses to specific comments from the reviewer(s) are listed below. Any corrections made are marked in Red. Please refer to those changes for more detailed information.
Reviewer 1
The submitted paper reports the adsorption of P, Pb and Cd on landfill leachate sludge-derived biochar. The adsorption mechanisms are discussed. It was proved that the removal of cations was mainly controlled by the mineralization induced by the released chemicals. However, at low concentration, it is said the uptake was mainly through physical adsorption, which was based on the results of simple modeling. I do not think it is accurate and convincing. The authors should setup more experiments and provide more proofs. Moreover, as regard to the mineralization, more details should be given, such as the change of pH and released ions in solution. Therefore, I recommend it to be resubmitted after a major revision.
Response:we have added the change of pH and released ions in solution to the newly submitted manuscript (Figure 2 Changes in ions concentrations and pH in wastewater at different initial pollutant concentrations absorbed by LLSDB600 (a: TP; b: Cd(II); c: Pb(II). LLSDB dosage of 6 g/L, adsorption temperature of 10 ℃ and Adsorption time of 2.0 hours) and a more specific description. (Line 199-209, Page 6, Line 310-312, Page 10)
As the reviewer's suggestion, the authors checked and corrected the grammatical errors in the article. Many modifications had been made, as highlighted in red to improve the readability of the text.

Reviewer 2 Report
In this manuscript, the authors investigate the mechanism of phosphorus and heavy metal absorption by landfill leachate sludge-derived biochar (LLSDB). Initially, the authors describe the preparation and characterization of LLSDB. Subsequently, they examine the dependence of the absorption rate of pollutants on factors such as pyrolysis temperature, absorption temperature, and dosage. Lastly, the authors investigate the kinetics and thermodynamics of the removal process and propose absorption mechanisms. While the results presented in this work are potentially valuable, there are several issues that need to be addressed to enhance the overall rigor of this manuscript before I can recommend its publication.
1. In the Introduction, the authors provide a detailed description of the motivation and importance of their work. However, a summary of the results, which is an essential component of the Introduction, is missing. I recommend that the authors highlight the significant findings of this study after the sentence "Findings of this study encouraged further investigation to comprehensively evaluate the application prospects of LLSDB."
2. Figure 1
(1) In Figure 1, the error bars are barely visible due to the large marker size. It would be preferable to use a smaller marker size or consider using a different marker shape, rather than the large triangle, to represent the data points. This adjustment would enhance the clarity and readability of the figure.
(2) The y-axis label of Figure 1b is "Absorption ratio," while the y-axis labels for Figures 1a and 1c are "Absorption rate." It is unclear whether these two terms refer to the same concept. If they do, I recommend using the same y-axis label for all three figures to ensure consistency and avoid confusion.
3. Figure 3
(1) The SEM images in Figure 3b require a higher resolution. The text within the images is currently illegible. To improve the clarity and readability, it is necessary to provide higher resolution versions of the SEM images, ensuring that the text is clearly visible and legible for readers.
(2) The authors state, "After adsorption, a large number of small particle clusters gathered on the surface of LLSDB600, filling most of the pores that originally existed on the surface, and there were a large number of small particles arranged densely on the LLSDB600’s surface." However, this observation is not evident from the SEM images shown in Figure II-IV. To clarify this point, it would be beneficial for the authors to clearly label the smaller particles in the SEM images. This would allow readers to visually identify and understand the presence of these particle clusters and their arrangement on the surface of LLSDB600.
4. On page 11, the authors utilize several models, including the Langmuir model, Freundlich model, Dubin Radushkevich (D-R) model, and Temkin model, to fit the experimental data. However, it is important for the authors to include references for these models. Without proper references, readers may not have a clear understanding of what these models entail. Providing appropriate citations for each model will allow readers to access the relevant literature and comprehend the theoretical background and assumptions underlying these models.
5. On page 18, the authors mention that "The morphological composition of each substance was calculated and drawn in the equilibrium system using HYDRA/MEDUSA software..." However, it would be helpful for the authors to provide a description of the calculation process involved and include references or additional information about the HYDRA/MEDUSA software. Currently, it is not at all clear to readers how the morphological compositions were computed using this software.
6. It is important to define abbreviations before their first use. In the abstract, the abbreviation "TP" is used without prior definition. In fact, "TP" is not defined until line 125 on page 3.
There are numerous grammatical errors throughout the manuscript, some of which may convey misinformation or greatly affect readability. I strongly recommend that the authors refine the English language used in the manuscript, potentially seeking assistance from a native English speaker, in order to improve its overall quality. Below are just the errors identified in the abstract:
"It was investigated on the effect of nutrients phosphorus, heavy metals (Pb(II), Cd(II)) adsorption removals in wastewater and the kinetics, the thermodynamics for adsorption by LLSDB." - This sentence is difficult to understand due to a grammatical error.
Additionally, right after this sentence, the word "it" should be capitalized.
I suggest thoroughly reviewing the manuscript for other grammatical errors and considering a comprehensive proofreading to enhance the clarity and readability of the text.
Author Response
We are very grateful to Reviewer for reviewing the paper so carefully.
We have completely revised the manuscript to clarify the points raised by the reviewer(s). The responses to specific comments from the reviewer(s) are listed below. Any corrections made are marked in Red. Please refer to those changes for more detailed information.
Reviewer 2
In this manuscript, the authors investigate the mechanism of phosphorus and heavy metal absorption by landfill leachate sludge-derived biochar (LLSDB). Initially, the authors describe the preparation and characterization of LLSDB. Subsequently, they examine the dependence of the absorption rate of pollutants on factors such as pyrolysis temperature, absorption temperature, and dosage. Lastly, the authors investigate the kinetics and thermodynamics of the removal process and propose absorption mechanisms. While the results presented in this work are potentially valuable, there are several issues that need to be addressed to enhance the overall rigor of this manuscript before I can recommend its publication.
Comments and Suggestions for Authors
- In the Introduction, the authors provide a detailed description of the motivation and importance of their work. However, a summary of the results, which is an essential component of the Introduction, is missing. I recommend that the authors highlight the significant findings of this study after the sentence "Findings of this study encouraged further investigation to comprehensively evaluate the application prospects of LLSDB."
Response: At the conclusion of the introduction, a summary of the experimental results has been included. (Line 92-93, Page 3)
- Figure 1
(1)In Figure 1, the error bars are barely visible due to the large marker size. It would be preferable to use a smaller marker size or consider using a different marker shape, rather than the large triangle, to represent the data points. This adjustment would enhance the clarity and readability of the figure.
(2)The y-axis label of Figure 1b is "Absorption ratio," while the y-axis labels for Figures 1a and 1c are "Absorption rate." It is unclear whether these two terms refer to the same concept. If they do, I recommend using the same y-axis label for all three figures to ensure consistency and avoid confusion.
Response: In Figure 1, the label size and shape were modified based on the editor’s suggestions, and the Y-axis was standardized across all three figures. This error was also corrected in the other diagrams. (Line 192-193, Page 6)
Since a new diagram was added after Figure 1, the order of the diagrams in the article has been changed, so I have chosen to use the new diagram order in subsequent replies.
3.Figure 3
(1)The SEM images in Figure 3b require a higher resolution. The text within the images is currently illegible. To improve the clarity and readability, it is necessary to provide higher resolution versions of the SEM images, ensuring that the text is clearly visible and legible for readers.
(2)The authors state, "After adsorption, a large number of small particle clusters gathered on the surface of LLSDB600, filling most of the pores that originally existed on the surface, and there were a large number of small particles arranged densely on the LLSDB600’s surface." However, this observation is not evident from the SEM images shown in Figure II-IV. To clarify this point, it would be beneficial for the authors to clearly label the smaller particles in the SEM images. This would allow readers to visually identify and understand the presence of these particle clusters and their arrangement on the surface of LLSDB600.
Response: Based on the editor’s suggestion, in Figure 4, the magnification scale in the SEM image has been labeled, the previous cropping of the mark in the original image has been corrected, and smaller particles are indicated in the image. (Line 324-325, Page 11)
4.On page 11, the authors utilize several models, including the Langmuir model, Freundlich model, Dubin Radushkevich (D-R) model, and Temkin model, to fit the experimental data. However, it is important for the authors to include references for these models. Without proper references, readers may not have a clear understanding of what these models entail. Providing appropriate citations for each model will allow readers to access the relevant literature and comprehend the theoretical background and assumptions underlying these models.
Response: In the model construction section, according to the editor's suggestion, the relevant references of the model are added to facilitate readers' understanding and reading. See references [37, 38, 40, 41] for details. (Line256, 264, 268, Page 8)
5.On page 18, the authors mention that "The morphological composition of each substance was calculated and drawn in the equilibrium system using HYDRA/MEDUSA software..." However, it would be helpful for the authors to provide a description of the calculation process involved and include references or additional information about the HYDRA/MEDUSA software. Currently, it is not at all clear to readers how the morphological compositions were computed using this software.
Response: References to related software have been added to the article, see [45,46] References for more information. (Line330-337, Page 11)
6.It is important to define abbreviations before their first use. In the abstract, the abbreviation "TP" is used without prior definition. In fact, "TP" is not defined until line 125 on page 3.
Response: As per the editor’s suggestion, abbreviations were defined before their first usage, and other similar instances in the article were carefully reviewed. (Line 18, Page 1)
Comments on the Quality of English Language
There are numerous grammatical errors throughout the manuscript, some of which may convey misinformation or greatly affect readability. I strongly recommend that the authors refine the English language used in the manuscript, potentially seeking assistance from a native English speaker, in order to improve its overall quality. Below are just the errors identified in the abstract:"It was investigated on the effect of nutrients phosphorus, heavy metals (Pb(II), Cd(II)) adsorption removals in wastewater and the kinetics, the thermodynamics for adsorption by LLSDB." - This sentence is difficult to understand due to a grammatical error.Additionally, right after this sentence, the word "it" should be capitalized.I suggest thoroughly reviewing the manuscript for other grammatical errors and considering a comprehensive proofreading to enhance the clarity and readability of the text.
Response: As per the editor’s suggestion, the author checked and corrected the grammatical errors in the article. Additionally, frequently used words were replaced with red-highlighted ones to improve the readability of the text. Additionally, the issues in the abstract have been modified (Line 15-17, Page 1)

Round 2
Reviewer 1 Report
The authors have done proper revision to the manuscript, and it can be published.
Minor revision is needed.
Author Response
We are very grateful to Reviewer for reviewing the paper so carefully.
Reviewer 2 Report
The revisions have addressed the majority of the issues identified in the original submission. However, I have a minor suggestion for the authors to consider: including a description of the green circles depicted in Figure 4, which represent the small particle clusters, in both the main text and the figure's caption. Currently, I am unable to locate any such description, and this omission may potentially confuse readers.
English is fine and it would be best if the language can be further polished.
Author Response
We are very grateful to Reviewer for reviewing the paper so carefully.
We have completely revised the manuscript to clarify the points raised by the reviewer(s). The responses to specific comments from the reviewer(s) are listed below. On the basis of the last revision, this revisions made to the manuscript should be marked up using the “Track Changes”. Please refer to those changes for more detailed information.
Reviewer 2
The revisions have addressed the majority of the issues identified in the original submission. However, I have a minor suggestion for the authors to consider: including a description of the green circles depicted in Figure 4, which represent the small particle clusters, in both the main text and the figure's caption. Currently, I am unable to locate any such description, and this omission may potentially confuse readers.
Response: The annotation for the green circle in Figure 4 has been added in both the figure and the text.
Meanwhile, we also made minor corrections to the sentences in the article to make them more coherent. We carefully reviewed the references and corrected any errors found(The errors in the references were not marked in the text).
